# Association of TNF-α (-308G/A) Gene Polymorphism with Changes in Circulating TNF-α Levels in Response to CPAP Treatment in Adults with Coronary Artery Disease and Obstructive Sleep Apnea

**DOI:** 10.3390/jcm12165325

**Published:** 2023-08-16

**Authors:** Yeliz Celik, Yüksel Peker, Tülay Yucel-Lindberg, Tilia Thelander, Afrouz Behboudi

**Affiliations:** 1Department of Pulmonary Medicine, Koc University School of Medicine, and Koc University Research Center for Translational Medicine (KUTTAM), Koc University, 34010 Istanbul, Turkey; yecelik@ku.edu.tr; 2Department of Molecular and Clinical Medicine, Sahlgrenska Academy, University of Gothenburg, 40530 Gothenburg, Sweden; 3Department of Clinical Sciences, Respiratory Medicine and Allergology, Faculty of Medicine, Lund University, 22185 Lund, Sweden; 4Division of Pulmonary, Allergy, and Critical Care Medicine, University of Pittsburgh School of Medicine, Pittsburgh, PA 15213, USA; 5Division of Sleep and Circadian Disorders, Brigham and Women’s Hospital, and Harvard Medical School, Boston, MA 02115, USA; 6Department of Dental Medicine, Karolinska Institute, 17177 Stockholm, Sweden; tulay.lindberg@ki.se; 7Division of Biomedicine, School of Heath Sciences, University of Skövde, 54128 Skövde, Sweden; tilia.thelander@gmail.com (T.T.); afrouz.behboudi@his.se (A.B.)

**Keywords:** coronary artery disease, obstructive sleep apnea, tumor necrosis factor

## Abstract

Rationale: We recently demonstrated that patients with coronary artery disease (CAD) and obstructive sleep apnea (OSA) carrying the tumor necrosis factor-alpha *(TNF-α) A* allele had increased circulating *TNF-α* levels compared with the ones carrying the *TNF-α G* allele. In the current study, we addressed the effect of *TNF-α* (-308G/A) gene polymorphism on circulating *TNF-α* levels following continuous positive airway pressure (CPAP) therapy. Methods: This study was a secondary analysis of the RICCADSA trial (NCT00519597) conducted in Sweden. CAD patients with OSA (apnea–hypopnea index) of ≥15 events/h and an Epworth Sleepiness Scale (ESS) score of <10 were randomized to CPAP or no-CPAP groups, and OSA patients with an ESS score of ≥10 were offered CPAP treatment. Blood samples were obtained at baseline and 12-month follow-up visits. *TNF-α* was measured by immunoassay (Luminex, R&D Systems). Genotyping of *TNF-α*-308G/A (single nucleotide polymorphism Rs1800629) was performed by polymerase chain reaction–restriction fragment length polymorphism. Results: In all, 239 participants (206 men and 33 women; mean age 64.9 (SD 7.7) years) with polymorphism data and circulating levels of *TNF-α* at baseline and 1-year follow-up visits were included. The median circulating *TNF-α* values fell in both groups between baseline and 12 months with no significant within- or between-group differences. In a multivariate linear regression model, a significant change in circulating *TNF-α* levels from baseline across the genotypes from GA to GA and GA to AA (standardized β-coefficient −0.129, 95% confidence interval (CI) −1.82; −0.12; *p* = 0.025) was observed in the entire cohort. The association was more pronounced among the individuals who were using the device for at least 4 h/night (n = 86; standardized β-coefficient −2.979 (95% CI −6.11; −1.21); *p* = 0.004)), whereas no significant association was found among the patients who were non-adherent or randomized to no-CPAP. The participants carrying the *TNF-α A* allele were less responsive to CPAP treatment regarding the decline in circulating *TNF-α* despite CPAP adherence (standardized β-coefficient −0.212, (95% CI −5.66; −1.01); *p* = 0.005). Conclusions: Our results suggest that *TNF-α* (-308G/A) gene polymorphism is associated with changes in circulating *TNF-α* levels in response to CPAP treatment in adults with CAD and OSA.

## 1. Introduction

Coronary artery disease (CAD) is associated with high mortality [1]. The traditionally recognized risk factors for CAD are age, male sex, hypertension, diabetes, and hyperlipidemia. It has also been proposed that the interaction between genetic and environmental factors influences the development of CAD [1,2,3].

Obstructive sleep apnea (OSA) is characterized by intermittent upper airway collapse during sleep, causing sleep fragmentation and intermittent hypoxia [4]. Almost 50% of CAD patients have OSA, and many of them do not report excessive daytime sleepiness (ESS), which is one of the cardinal symptoms of OSA [5]. Individuals with OSA have been reported to have an increased risk of incident CAD compared with adults without OSA [6].

Vascular inflammation plays a key role in the development of atherosclerotic plaques and CAD [7]. It has also been suggested that circulating levels of inflammatory markers can predict future cardiovascular events [8,9]. Elevated levels of high-sensitivity C-reactive protein (hs-CRP), interleukin (IL)-6, and tumor necrosis factor (TNF)-α have been reported in adults with OSA [10,11]. Treatment of OSA with continuous positive airway pressure (CPAP) has been suggested to normalize the levels of circulating inflammatory markers, supporting the link between systemic inflammation and OSA [12]. It has also been proposed that inflammation can be a predisposing factor for OSA [13,14,15], not just a consequence of OSA.

*TNF-α* is a pro-inflammatory cytokine that is important for the immune system and plays a notable role in the development of autoimmune and infectious diseases as well as atherosclerosis and CAD [16]. *TNF-α* also plays a crucial role in sleep regulation [17]. Many OSA patients have elevated levels of circulating *TNF-α* [18]. Existing data also suggest that genetic and environmental factors are involved in the development of OSA [19], and *TNF-α* has received special attention in this context [17,20]. There is an SNP (Rs1800629) in the promoter region of the *TNF-α* (position 308G/A); allele A at this position (*TNF-*α-308A) is suggested to be associated with a higher occurrence of OSA [21] as well as with the severity of this disorder [18,22,23,24,25]. There are also reports concerning the association of *TNF-α*-308G/A (rs1800629) polymorphism with the risk of many diseases, such as allograft rejection [26], asthma [27], chronic obstructive pulmonary disease [28], ischemic stroke [29], rheumatoid arthritis [30], and systemic lupus erythematosus [31]. An association between the *TNF-*α-308A allele and obesity has also been reported [17,21], whereas conflicting results have been reported regarding the relationship between *TNF-α*-308G/A polymorphism and CAD. One study suggested that *TNF-α*-308G/A polymorphism is associated with ST-elevation myocardial infarction and high plasma levels of biochemical ischemia markers [32], and a meta-analysis demonstrated a significant association between *TNF-α*-308G/A and the risk of acute myocardial infarction [33]. On the other hand, a recent meta-analysis showed no significant association [34].

We recently demonstrated that patients with CAD and OSA carrying the *TNF-α A* allele had increased circulating *TNF-α* levels compared with the ones carrying the *TNF-α G* allele [35] in the “Randomized Intervention with CPAP in CAD and OSA” (RICCADSA) cohort [36]. In the current study, we addressed the role of *TNF-α* (-308G/A) gene polymorphism on circulating *TNF-α* levels in response to 12 months of CPAP therapy.

## 2. Materials and Methods

### 2.1. Study Participants

The methodology of the main RICCADSA trial was described elsewhere [36]. In total, 511 CAD patients who underwent percutaneous coronary intervention (PCI) or coronary artery bypass grafting (CABG) in Skaraborg County of West Götaland, Sweden, were included in the RICCADSA trial between 2005 and 2010 (Figure 1). The participants with OSA, defined as an apnea–hypopnea index (AHI) of ≥15/h, on the home sleep apnea test (HSAT) at screening and an Epworth Sleepiness Scale (ESS) score of <10 were randomized to CPAP or no-CPAP groups. Patients with ESS scores of ≥10 were categorized as having excessive daytime sleepiness (EDS) and were offered CPAP treatment. The CAD patients with AHI < 5/h were categorized as no-OSA in the main protocol. For the genetic analysis, blood samples were collected at the final visit in 2012/2013 from 384 eligible participants, and 239 patients with OSA were included as the final study population for the current *TNF-α*-308G/A polymorphism study to evaluate the changes in circulating *TNF-α* levels from baseline to 12 months after CPAP treatment (Figure 1).

### 2.2. Study Oversight

The study protocol was approved by the Ethics Committee of the Medical Faculty of the University of Gothenburg (approval nr 207-05, 09.13.2005; amendment T744-10, 11.26.2010; amendment T512-11, 06.16.2011; additional approval for the molecular analysis, approval nr 814-17, 11.21.2017). Written informed consent was obtained from all participants. The main RICCADSA trial was registered with ClinicalTrials.gov (NCT 00519597).

### 2.3. Sleep Studies

The Embletta^®^ Portable Digital System device (Embla, Broomfield, CO, USA) was used for the HSATs [36]. Apnea was defined as at least a 90% cessation of airflow, and hypopnea was defined as at least a 50% reduction in nasal pressure amplitude and/or thoracoabdominal movement for at least 10 s, following the Chicago criteria [37]. The total number of significant drops in SpO_2_ exceeding 4% from the immediately preceding baseline was also recorded, and the oxygen desaturation index (ODI) was determined as the number of significant desaturations per hour.

### 2.4. Epworth Sleepiness Scale

The ESS [38] was assessed to measure subjective daytime sleepiness. The ESS has eight items asking about the risk of dozing off under 8 different situations, and a score of at least 10 out of 24 was defined as EDS.

### 2.5. Comorbidities

Demographics, smoking habits, and medical history of the study cohort were obtained from the medical records. Individuals with a body mass index (BMI) of ≥30 kg/m^2^ were defined as obese, and abdominal obesity was defined as a waist-to-hip ratio (WHR) of ≥0.9 for men and a WHR of ≥0.8 for women [39].

### 2.6. TNF-α Circulating Concentration

All blood samples were collected in the morning (07:00–08.00 am) after overnight fasting using EDTA (ethylenediaminetetraacetic acid) tubes, as previously described [40]. The tubes underwent centrifugation, and the resulting plasma/serum samples were divided into aliquots and subsequently stored at −70 °C until analysis. Circulating *TNF-α* levels were measured in the plasma samples (undiluted) using commercially available MILLIPLEX MAP (based on Luminex technology) human adipokine assay kits according to the manufacturer’s instructions (Merck Millipore, Burlington, MA, USA). The assay sensitivities (minimum detectable levels) for *TNF-α* were 0.14 pg/mL, and all samples exhibited levels within the standard curve, covering a spectrum of 0 to 10,000 pg/mL. The intra-assay variability ranged from 1.4% to 7.9%, while the inter-assay variability was below 21% for the assessment of *TNF-α* concentrations. These values were calculated from the mean of the percentage coefficient of variability from multiple reportable results across two different concentrations of the samples in one experiment or from two results each for two different concentrations of samples across several different experiments.

### 2.7. TNF-α Promotor -308G/A (Rs1800629) SNA Genotyping

As previously described in detail [35], genomic DNA was isolated from whole blood samples collected in EDTA-coated tubes using the PAXgene Blood DNA Kit (PreAnalytiX; Qiagen). The quality and concentration of DNA samples were determined using a nanodrop photometer (NanoDrop 2000; Thermo Scientific, Waltham, MA, USA), and DNA samples were stored at −80 degrees. *TNF-α* promoter −308A/G (Rs1800629) genotyping analysis was performed by polymerase chain reaction–restriction fragment length polymorphism (PCR–RFLP), as previously described [35].

### 2.8. Statistical Analysis

For descriptive statistics, variables were reported as medians with interquartile ranges (IQR) for continuous variables and as percentages for categorical variables. The Shapiro–Wilk test was used to test the normality assumption of the current data for all variables. Between-group differences stratified by CPAP allocation and CPAP usage in baseline characteristics, as well as changes from baseline in circulating *TNF-α* levels, were tested by the Mann–Whitney test for continuous variables and the Chi-square test for categorical data. Within-group differences in changes from baseline in circulating *TNF-α* levels were tested by the Wilcoxon signed-rank test. A univariate linear regression analysis was performed to test the association between the change from baseline to the 12-month follow-up in circulating *TNF-α* levels and age, sex, ESS, BMI, WHR, AHI, ODI, OSA, and comorbidities, as well as *TNF-α* genotypes (coded as GG = 0, GA = 1, and AA = 2) and *TNF-α* alleles (coded as G = 0, and A = 1), respectively. Multivariate models included the same significant covariates as the univariate analysis as well as the variables of age, BMI, and sex in order to align with the recent guidelines [41]. All statistical tests were two-sided, odds ratios (ORs) with 95% confidence interval (CI) were reported, and a *p*-value of <0.05 was considered significant. Statistical analyses were performed using SPSS^®^ 28.0 for Windows^®^ (SPSS Inc., Chicago, IL, USA).

## 3. Results

The study population consisted of 239 participants (mean age 64.9 ± 7.7 years; male, 86%). As presented in Table 1, patients allocated to the no-CPAP group were slightly older and less sleepy, and the proportion of individuals with diabetes at baseline was lower than that among the patients allocated to CPAP treatment. The circulating levels of *TNF-α* at baseline did not differ significantly between the groups.

As illustrated in Figure 2A, *TNF-α*-GG was the most prevalent genotype in both groups, whereas *TNF-α*-AA in the CPAP group and *TNF-α*-GA in the no-CPAP group were the least frequent ones, respectively.

The median circulating levels of *TNF-α* decreased from 4.87 (3.43–6.99) pg/mL to 4.62 (3.59–6.59) pg/mL in patients allocated to the CPAP group (*p* = 0.549) and from 5.15 (3.92–6.54) pg/mL to 4.50 (3.64–7.11) pg/mL in patients allocated to the no-CPAP group (*p* = 0.665), with no significant between-group differences in the magnitude of change from baseline.

When analyzing the study population after stratifying by CPAP usage, the baseline characteristics did not differ significantly, except for ESS scores and the proportion of individuals with baseline EDS, which were higher among patients who used the device for at least 4 h/night during the first 12 months (Table 2).

As illustrated in Figure 3A, *TNF-α*-GG was the most prevalent genotype and *TNF-α*-AA the least frequent one in both CPAP usage groups.

As illustrated in Figure 4, the median circulating levels of *TNF-α* decreased from 4.84 (3.48–7.53) pg/mL to 4.72 (3.63–7.20) pg/mL in patients who used the device for at least 4 h/night (*p* = 0.577) and from 5.24 (3.59–6.85) pg/mL to 4.51 (3.50–6.75) pg/mL in patients allocated to the no-CPAP group or who used the device for less than 4 h/night (*p* = 0.199), with no significant between-group differences in the magnitude of change from baseline.

In a multivariate linear regression model, a significant decline in the change from baseline in circulating *TNF-α* levels across the genotypes from GG to GA and GA to AA was observed in the entire cohort (Table 3). The association was more pronounced among individuals who were using the device for at least 4 h/night, whereas no significant association was found among the patients who were non-adherent or randomized to the no-CPAP group. ESS scores at baseline tended to be inversely correlated with the change in circulating *TNF-α* levels from baseline to 12 months in the entire cohort (Table 3).

As shown in Table 4, the participants carrying the *TNF-α A* allele were less responsive to CPAP treatment regarding the decline in circulating *TNF-α* levels despite CPAP adherence. Baseline AHI was also inversely correlated with a decline in the change from baseline in circulating *TNF-α* levels among patients who were adherent to CPAP. No significant changes were observed among patients who were randomized to the no-CPAP group or those who were using the device for less than 4 h/night, except for baseline AHI, which was associated with the change in circulating *TNF-α* levels (Table 4).

## 4. Discussion

In the current revascularized CAD cohort with OSA, *TNF-α*-308G/A gene polymorphism was significantly correlated with the change in circulating *TNF-α* levels from baseline in response to 12 months of CPAP treatment, independent of age, sex, BMI, baseline AHI, ESS, and diabetes. The participants carrying the *TNF-α* A allele were less responsive to CPAP treatment in terms of the decline in circulating *TNF-α* levels despite adequate CPAP adherence levels.

To the best of our knowledge, this is the first study to address the association of *TNF-α*-308G/A polymorphism with change in circulating *TNF-α* levels in response to the alleviation of OSA with CPAP treatment in a Swedish cardiac population. Previous studies have suggested an association between the *TNF-α* -308A allele and OSA susceptibility in a British population [24] as well as in an obese Asian Indian population [22], whereas neutral results were reported in a Polish cohort [21] and a Turkish cohort [42]. Notwithstanding, two meta-analyses have supported the significant association between *TNF-α* -308G/A polymorphism and OSA [43,44].

In our entire cohort including adults without OSA as a control group, we found no significant difference between CAD patients with vs. without OSA in the frequency of *TNF-α*-308G/A polymorphism or *TNF-α* -308 alleles [35]. We found a similar -308A allele frequency in the no-OSA group, and, as interpreted in the previous report, this might be due to the confounding effect of other comorbidities, such as obesity, hypertension, and diabetes mellitus, given that individuals without OSA were not healthy controls [27]. Moreover, CAD *per se* is an inflammatory condition mediated by the activity of pro-inflammatory cytokines, including *TNF-α* [45]. The effect of *TNF-α* gene polymorphism on CAD pathogenesis has also been investigated previously, and *TNF-α*-308G/A polymorphism has been suggested to be involved in CAD development [45,46], whereas others reported no evidence for such an association [47]. Additionally, a recent meta-analysis suggested no significant relationship between *TNF-α*-308G/A polymorphism and the development of CAD [34].

Circulating levels of inflammatory markers predict future cardiovascular events in the general population [8] as well as in cardiac populations [9], and *TNF-α* levels are elevated in individuals with OSA compared with controls [25,48]. It has also been argued that inflammation can be a predisposing factor for OSA [13,14,15]; thus, this association could be bidirectional. In our first study on the effect of CPAP on inflammatory markers, including hs-CRP, IL-6, IL-8, and *TNF-α*, in the RICCADSA cohort, only IL-6 levels decreased after one year, both in the CPAP and no-CPAP arms [49]. This was probably indicative of a natural improvement in cardiac disease rather than the effect of CPAP treatment per se. We also demonstrated that patients with CAD and OSA carrying the *TNF-α A* allele had increased circulating *TNF-α* levels compared with the ones carrying the *TNF-α G* allele [35]. The *TNF-α*-308A allele is known to promote a two-fold increase in TNF transcription activity [50]. *TNF-α* is a mediator of the sleep regulatory system, and the fragmented sleep pattern associated with OSA is believed to increase circulatory levels of *TNF-α* [17]. Our current results clearly support a recent review of OSA heterogeneity regarding cardiovascular morbidities [1], suggesting that the response to CPAP treatment is modulated by genetic mechanisms, namely, in the current report, by *TNF-α* (-308G/A) gene polymorphism regarding the change in circulating levels of TNF-𝛼 from baseline. In other words, individuals with CAD and OSA carrying the *TNF-α A* allele seem to have an increased risk of elevated levels of circulating *TNF-α*, and the increased inflammatory activity is less likely to normalize despite CPAP use for at least 4 h/night. Whether or not those individuals should use the device even longer in order to reduce the levels of circulating *TNF-α* levels warrants further research. The clinical implications of our findings may also include that *TNF-α*-308G/A genotyping together with the analysis of *TNF-α* levels can be used for the prognostic evaluation of patients with CAD and concomitant OSA. For instance, patients carrying the A allele and higher levels of *TNF-α* could be included in a tighter follow-up scheme compared with those carrying the G allele.

We should acknowledge certain limitations. As also stated in the previous report [35], the power estimate for the entire RICCADSA cohort was conducted for the primary outcome and not for the secondary outcomes assessed in this study. Moreover, our results are limited to a Swedish CAD cohort and thus are not generalizable to adults with OSA in the general population or sleep clinic cohorts in other regions. Additionally, the follow-up period was relatively short in the context of the association of changes in circulating TNF-𝛼 levels at 12 months with long-term adverse outcomes. Finally, our results are limited to *TNF-α* (-308G/A) (rs1800629) polymorphism and not the *TNF-α* (-308G/A) (rs3611525) polymorphism. However, *TNF-α*-308G/A (rs1800629) and *TNF-α*-238G/A (rs361525) are located very close to each other and are hence tightly linked. In fact, in our sequencing results, we noticed that polymorphism in one 100% mirrored the other, and for the sake of simplicity, we chose to focus on and present only one of these SNPs. Accordingly, we propose that the *TNF-α*-308G/A (rs1800629) polymorphism analysis results can be extrapolated to *TNF-α*-238G/A (rs361525) SNP due to their tight linkage.

## 5. Conclusions

We conclude that *TNF-α*-308G/A gene polymorphism was significantly correlated with the change in the circulating *TNF-α* levels from baseline in response to 12 months of CPAP treatment in this revascularized Swedish CAD cohort, independent of age, sex, BMI, baseline AHI, ESS, and diabetes. The participants carrying the *TNF-α* A allele were less responsive to CPAP treatment in terms of the decline in circulating *TNF-α* levels despite adequate CPAP adherence levels. Further prospective studies in larger cohorts and different geographical locations are warranted in order to better clarify whether the combined gentrifying and protein level analysis can be used as a prognostic biomarker for improved clinical follow-up of patients with CAD and concomitant OSA.

## Figures and Tables

**Figure 1 jcm-12-05325-f001:**
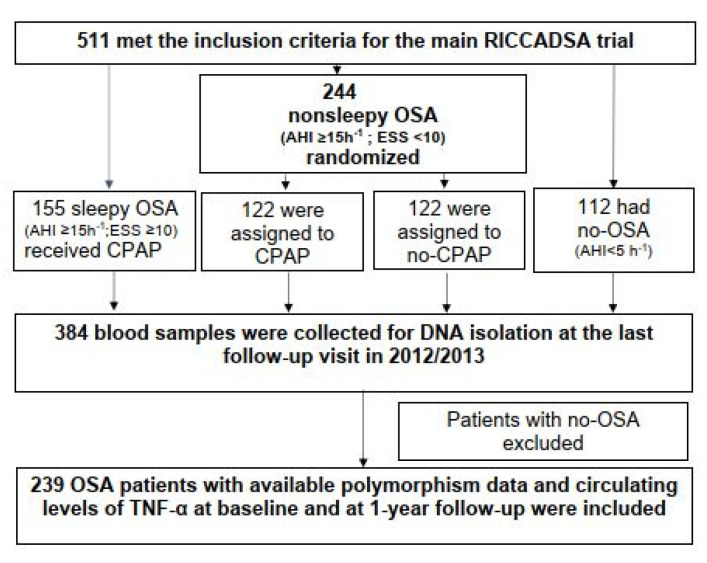
Analytic sample of the study population. Abbreviations: AHI, apneahypopnea index; CPAP, continuous positive airway pressure; ESS, Epworth Sleepiness Scale; OSA, obstructive sleep apnea; RICCADSA, Randomized Intervention with CPAP in Coronary Artery Disease and Sleep Apnea; TNF, tumor necrosis factor.

**Figure 2 jcm-12-05325-f002:**
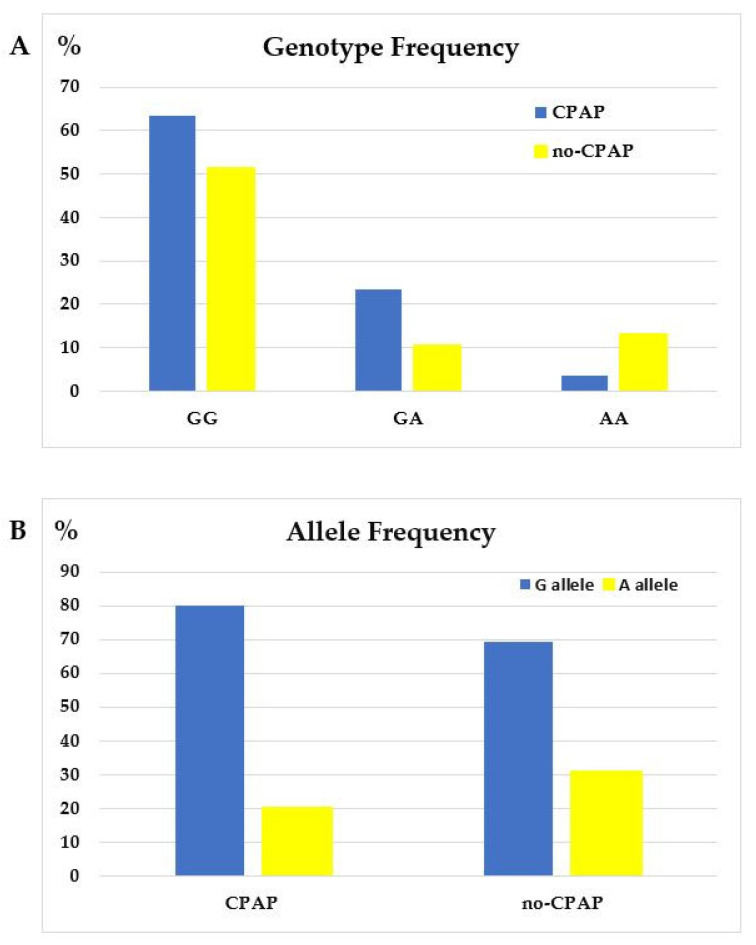
(**A**) Genotype frequency of *TNF-α*-308G/A promoter polymorphism and (**B**) allele frequency of *TNF-α*-308G/A promoter polymorphism in OSA patients allocated to CPAP vs. no-CPAP groups.

**Figure 3 jcm-12-05325-f003:**
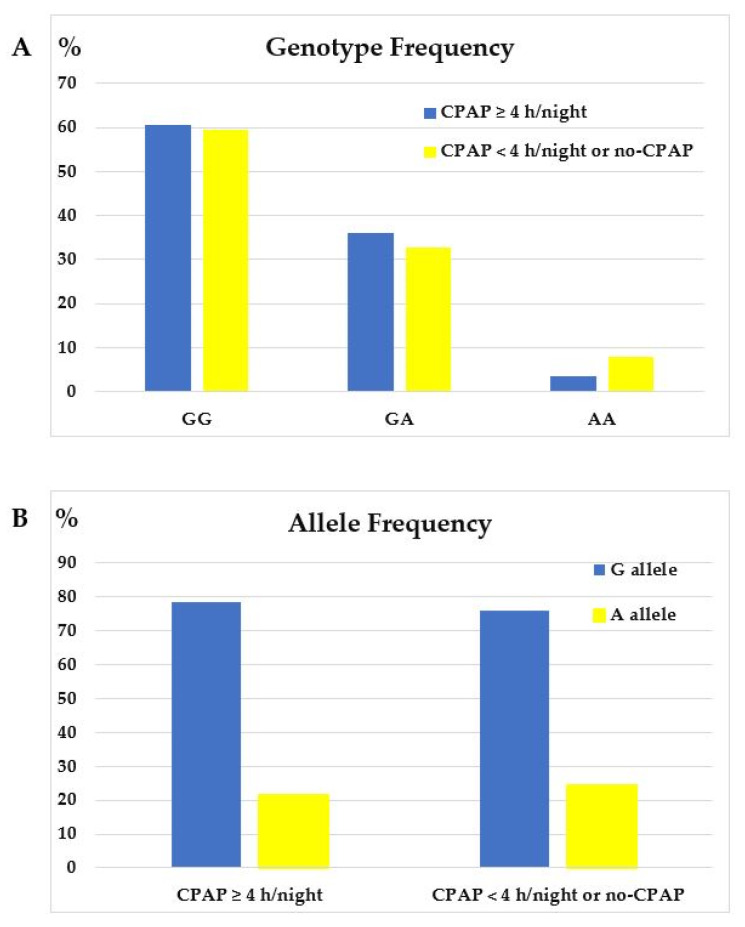
(**A**) Genotype frequency of *TNF-α*-308G/A promoter polymorphism and (**B**) allele frequency of *TNF-α*-308G/A promoter polymorphism in OSA patients stratified by CPAP usage.

**Figure 4 jcm-12-05325-f004:**
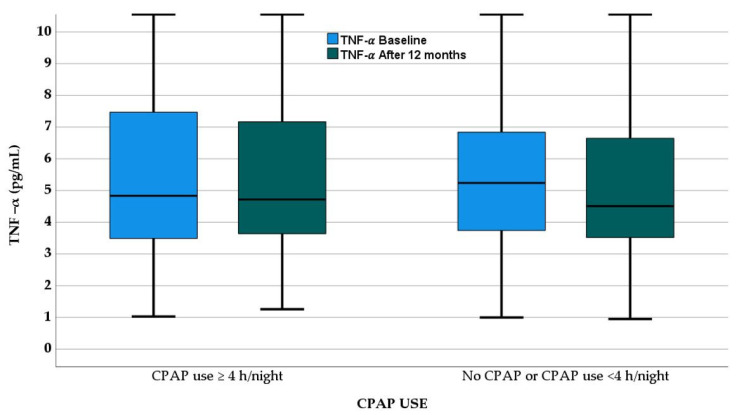
Circulating *TNF-α* levels at baseline and after 12 months of CPAP therapy in OSA patients stratified by CPAP usage categories.

**Table 1 jcm-12-05325-t001:** Baseline demographic and clinical characteristics of the OSA patients allocated to CPAP vs. no-CPAP groups.

	OSA on CPAP n = 169	OSA no-CPAP n = 70	*p*-Value
Age, yrs	64.1 (59.8–69.3)	67.4 (62.7–72.4)	0.019
Male sex, %	86.4	85.7	0.890
BMI, kg/m^2^	28.3 (25.9–31.1)	(28.7 (26.2–30.0)	0.548
Obesity, %	32.0	24.3	0.238
WHR	0.96 (0.93–1.00)	0.96 (0.91–0.99)	0.153
Abdominal obesity, %	93.2	95.7	0.470
Current smoking, %	16.0	15.7	0.960
ESS score	10.0 (6.0–12.0)	6.0 (4.0–7.0)	<0.001
EDS (ESS score ≥ 10), %	56.8	0.0	<0.001
AHI, events/h	27.7 (18.7–39.2)	22.9 (17.8–35.7)	0.128
ODI, events/h	15.7 (9.4–24.8)	12.6 (7.2–22.9)	0.101
Hypertension	60.9	54.3	0.341
AMI at baseline	54.4	44.3	0.153
Lung disease, %	5.3	4.3	0.738
Diabetes, %	26.0	12.9	0.026
Stroke, %	4.8	10.1	0.121
Plasma *TNF-α* (pg/mL)	4.87 (3.43–6.99)	5.15 (3.92–6.54)	0.856

Continuous data are presented as median and 25–75% quartiles. Categorical data are presented as percentages. Abbreviations: AHI = Apnea–Hypopnea Index; BMI = Body Mass Index; EDS = Excessive Daytime Sleepiness (ESS score ≥ 10); ESS = Epworth Sleepiness Scale; *TNF-α* = tumor necrosis factor-alpha; ODI = Oxygen Desaturation Index; OSA = Obstructive Sleep Apnea; WHR = Waist–Hip Ratio.

**Table 2 jcm-12-05325-t002:** Baseline demographic and clinical characteristics of the OSA patients stratified by CPAP usage.

	CPAP ≥ 4 h/Night (Adherent) n = 86	CPAP < 4 h/Night or no CPAP n = 153	*p*-Value
Age, yrs	64.4 (60.1–70.6)	65.2 (59.9–70.6)	0.602
Male sex, %	84.9	86.9	0.660
BMI, kg/m^2^	28.2 (25.7–31.1)	28.7 (26.2–30.2)	0.875
Obesity, %	32.6	28.1	0.470
WHR	0.96 (0.92–1.02)	0.97 (0.93–1.01)	0.609
Abdominal obesity, %	92.8	94.6	0.586
Current smoking, %	12.8	17.6	0.324
ESS score	10.0 (6.0–11.0)	7.0 (4.0–10.0)	0.007
EDS (ESS score ≥ 10), %	53.5	32.7	0.002
AHI, events/h	28.2 (18.3–40.1)	25.3 (18.6–36.2)	0.293
ODI, events/h	17.2 (10.1–25.7)	14.2 (7.9–23.1)	0.052
Hypertension	60.5	58.2	0.729
AMI at baseline	54.7	49.7	0.460
Lung disease, %	5.3	4.3	0.738
Diabetes, %	27.9	19.0	0.110
Stroke, %	5.8	4.6	0.674
Plasma *TNF-α* (pg/mL)	4.87 (3.48–7.53)	5.24 (3.59–6.85)	0.631

Continuous data are presented as median and 25–75% quartiles. Categorical data are presented as percentages. Abbreviations: AHI = Apnea–Hypopnea Index; BMI = Body Mass Index; EDS= Excessive Daytime Sleepiness (ESS score ≥ 10); ESS = Epworth Sleepiness Scale; *TNF-α* = tumor necrosis factor-alpha; ODI = Oxygen Desaturation Index; OSA = Obstructive Sleep Apnea; WHR = Waist–Hip Ratio.

**Table 3 jcm-12-05325-t003:** Regression analyses of the association of the *TNF-α* genotypes with change in circulating *TNF-α* levels from baseline, adjusted for the confounding variables in CAD patients with OSA (entire cohort and subgroups based on the CPAP use).

		Standardized	95% Confidence Interval for		*p*-Values
		Coefficients β	Lower Bound	Upper Bound	
Entire Cohort	Genotypes *	−0.129	−1.82	−0.12	0.025
	Age	0.018	−0.05	0.07	0.762
	Male sex	−0.035	−1.91	1.00	0.540
	BMI	−0.005	−0.15	0.14	0.942
	AHI	−0.035	−0.04	0.02	0.573
	ESS	−0.115	−0.28	0.00	0.056
	Diabetes	0.017	−1.13	1.53	0.768
CPAP use ≥4 h/night	Genotypes *	−2.979	−6.11	−1.21	0.004
	Age	−0.057	−0.25	0.15	0.607
	Male sex	0.128	−1.61	6.21	0.246
	BMI	0.029	−0.29	0.38	0.803
	AHI	−0.183	−0.16	0.01	0.096
	ESS	−0.078	−0.46	0.22	0.481
	Diabetes	0.006	−3.03	3.21	0.955
CPAP use <4 h/night or no-CPAP	Genotypes *	−0.019	−0.93	0.73	0.816
	Age	0.026	−0.58	0.79	0.765
	Male sex	−0.079	−2.36	0.84	0.349
	BMI	−0.078	−0.23	0.09	0.368
	AHI	0.124	−0.01	0.07	0.149
	ESS	−0.055	−0.19	0.10	0.507
	Diabetes	−0.055	−1.82	0.91	0.509

Abbreviations: AHI = Apnea–Hypopnea Index; BMI = Body Mass Index; CAD = Coronary Artery Disease; CPAP = Continuous Positive Airway Pressure; ESS = Epworth Sleepiness Scale; OSA = Obstructive Sleep Apnea. * GG = 0, GA = 1, AA = 2.

**Table 4 jcm-12-05325-t004:** Regression analyses of the association of the *TNF-α* A allele with change in circulating *TNF-α* levels from baseline, adjusted for the confounding variables in CAD patients with concomitant OSA.

		Standardized	95% Confidence Interval for		*p*-Values
		Coefficients β	Lower Bound	Upper Bound	
Entire Cohort	*TNF-α* A Allele	−0.098	−2.08	−0.08	0.034
	Age	0.020	−0.04	0.07	0.676
	Male sex	0.027	−0.88	1.60	0.568
	BMI	−0.009	−0.13	0.11	0.858
	AHI	−0.060	−0.05	0.10	0.202
	CPAP h/night	0.011	−0.14	0.18	0.827
	ESS	−0.061	−0.19	0.41	0.208
	Diabetes	−0.017	−1.24	0.85	0.714
CPAP use ≥4 h/night	*TNF-α* A Allele	−0.212	−5.66	−1.01	0.005
	Age	−0.038	−0.18	0.11	0.627
	Male sex	0.102	−0.90	4.58	0.187
	BMI	0.018	−0.21	0.26	0.823
	AHI	−0.189	−0.14	−0.02	0.015
	ESS	−0.061	−0.34	0.15	0.434
	Diabetes	0.012	−2.03	2.38	0.876
CPAP use <4 h/night or no-CPAP	*TNF-α* A Allele	−0.014	−0.96	0.75	0.805
	Age	0.025	−0.04	0.06	0.670
	Male sex	−0.079	−1.87	0.35	0.180
	BMI	−0.078	−0.18	0.04	0.197
	AHI	0.123	−0.01	0.06	0.039
	ESS	−0.055	−0.15	0.05	0.347
	Diabetes	−0.055	−1.40	0.49	0.346

Abbreviations: AHI = Apnea–Hypopnea Index; BMI = Body Mass Index; CAD = Coronary Artery Disease; CPAP = Continuous Positive Airway Pressure; ESS = Epworth Sleepiness Scale; OSA = Obstructive Sleep Apnea.

## Data Availability

Individual participant data that underlie the results reported in this article can be obtained by contacting the corresponding author, yuksel.peker@lungall.gu.se.

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
