# Peer review of "Association of TNF-α (-308G/A) Gene Polymorphism with Changes in Circulating TNF-α Levels in Response to CPAP Treatment in Adults with Coronary Artery Disease and Obstructive Sleep Apnea"

_jcm, 2023, doi:10.3390/jcm12165325_

Round 1

Reviewer 1 Report

Dear authors, I have studied with great interest the manuscript "Association of TNF-α (-308G/A) Gene Polymorphism with Changes in Circulating TNF-α Levels in Response to CPAP Treatment in Adults with Coronary Artery Disease and Obstructive Sleep Apnea"

The article submitted for consideration has scientific novelty, the study is well planned, materials and methods are clearly described, ethical control is used, correct methods of statistical processing are used.

The relevance lies in the fact that OSAS is widespread and has a number of common pathogenetic links with CVD: associations of OSAS with hypertension, cardiac arrhythmias and conduction disturbances, and an increased risk of sudden cardiac death at night have been established. In recent years, more and more data have appeared in the literature on the important role of inflammation in the pathogenesis of OSAS. In separate works, the authors point to an increased level of a number of proinflammatory cytokines in OSAS: C-reactive protein (CRP), tumor necrosis factor alpha (TNF-alpha), interleukin-6 (IL-6). The pathogenetic role of inflammatory mechanisms increases with the development of comorbid cardiovascular pathology and the formation of HF. However, there are unresolved questions about risk factors, the choice of biomarkers that are supposed to be involved in the pathogenesis of both OSA and cardiovascular diseases, as well as the effectiveness of response to treatment, in particular, to CPAP Treatment. Therefore, further research is needed in this direction.

The authors found that the effectiveness of CPAP Treatment in patients with very high cardiovascular risk who underwent myocardial revascularization, with SOAP, despite adherence to CPAP, is associated with polymorphism and does not depend on age, gender, BMI, presence of DM, and baseline levels of AHI and ESS.

Strengths of the study: specificity of the sample (patients with very high cardiovascular risk who underwent myocardial revascularization); consistent continuation of research in this area (Effect of Positive Airway Pressure on Cardiovas-391 cular Outcomes in Coronary Artery Disease Patients with Nonsleepy Obstructive Sleep Apnea. The RICCADSA Random-392 ized Controlled Trial); the paucity of studies devoted to the study of the contribution of pro-inflammatory genetic biomarkers to the effectiveness of OSA treatment using CPAP therapy; high potential for practical use of the results.

However, the authors should discuss the limitations of the study: relatively short follow-up period, results limited to only the Swedish cohort of patients

Reviewer 2 Report

The article is baffling me in the assessment.  Sections from introduction, material and methods, results and discussion were taken exactly from a previous study to the same authors and included in the current study under evaluation. 

I am also confused to consider this situation as a plagiarism or not?

I suggest for the authors to change  to change the style of their article into "Letter to the editor" with short but precise refer to their previous results and what is the difference in their new findings .

I can understand that the new findings are extension to your previous study already published. But this id not an excuse to repeat many of your previous paper to make a new readable paper

Reviewer 3 Report

Although the authors’ findings that TNF-α (-308G/A) gene polymorphism (re1800629) determines the changes in circulating TNF-α levels in response to continuous positive airway pressure (CPAP) treatment in adults with coronary artery disease (CAD) and obstructive sleep apnea (OSA) are interesting, numbers of points need clarifying and certain statements require further justification. These are given below.

1.      There are lots of reports concerning the association of TNF-α-308 G>A (rs1800629) polymorphism with the risk of many diseases such as allograft rejection (Pawlik, A. et al. Transplant. Proc. 2005, 37, 2041-2043), asthma (Aoki, T. et al. J. Hum. Genet. 2006, 51, 677-685; Wu, H. et al. Environ. Health Perspect. 2007, 115, 616-622; Munthe-Kaas, M.C. et al. Allergy 2007, 62, 991-998), chronic obstructive pulmonary disease (Zhang, S. et al. Respirology 2011, 16,107-115), Crohn’s disease (Ferreira, A.C. et al. Inflamm. Bowel Dis. 2005, 11, 331-339), exfoliation glaucoma (Mossböck, G. et al. Mol. Vis. 2019, 15, 518-522), Graves’ disease (Li, N. et al. Clin. Biochem. 2008, 41, 881-886), heart disease (Antonicelli, R. et al. Coron. Artery Dis. 2005, 16, 489-493; Settin, A. et al. Pediatr. Cardiol. 2007, 28,363-371; Hua, X.P. et al. Genet. Mol. Res. 2016, 15, gmr.15037292), ischemic stroke (Song, D. & Cheng, D. Genet. Test Mol. Biomarkers 2017, 21, 10-16), leprosy (Vejbaesya, S. et al. J. Med. Assoc. Thai. 2007, 90, 1188-1192), liver disease (Jeng, J.-E. et al. Neoplasia 2007, 9, 987-992), lymphoma (Jrad, B.B.H. et al. Eur. J. Haematol. 2007, 78,117-122; Cerhan, J.R. et al. Cancer Epidemiol. Biomarkers Prev. 2008, 17, 3161-3169; Wang, S.S. et al. Cancer Res.2006, 66, 9771-9780), susceptibility of Mediterranean spotted fever (Forte, G.I. et al. Clin Vaccine Immunol. 2009,16, 811-815), migraine (Schürks, M. et al. J. Pain 2009, 10, 759-766), multiple sclerosis (Risti´c, S. et al. Eur. Neurol. 2007, 57, 203-207), nasal polyps (Erbek, S. et al. Arch. Otolaryngol. Head Neck Surg. 2007, 133, 705-709; Bernstein, J.M. et alLaryngoscope 2009, 119, 1258-1264), psoriasis (Nedoszytko, N. et al. Br. J. Dermatol. 2007,157, 165-167; Balding, J. et al. Arthritis Rheum. 2003, 48, 1408-1413), rheumatoid arthritis (O’Rielly, D.D. et al.Pharmacogenomics J. 2009, 9, 161-167), sarcoidosis (Medica, I. et al. J. Hum. Genet. 2007, 52, 836-847), susceptibility of sepsis (Menges, T. et al. Crit. Care Med. 2008, 36, 1456-1462; Teuffel, O. et al. Crit. Care Med.2010, 38, 276-282), systemic lupus erythematosus (Guarnizo-Zuccardi, P. et al. Tissue Antigens 2007, 70, 376-382; Lee, Y.H. et al. Eur. J. Hum. Genet. 2006, 14, 364-371), and severity of sleep disturbance and morning fatigue (Aouizerat, B.E. et al. Biol. Res. Nurs. 2009, 11, 27-41). Therefore, why the TNF-α-308 G>A (rs1800629) polymorphism affected the changes in circulating TNF-α levels in response to CPAP treatment should be considered and discussed in relations of the previous reports. 

2.      In Table 1, diabetes ratio was significantly different between CPAP and no-CPAP groups. Was diabetes prevalence a confounding factor?

3.      Not only TNF-α-308 G>A (rs1800629) polymorphism but also TNF-α-238 G>A (rs361525) were reported (Li, C. et al. J, Invest. Dermatol 2007, 127, 1886-1892; Santos, N.C.D. et al. Cytokine 2020, 134, 155183). How the effect of rs361525 on the levels of TNF-α between CPAP and no-CPAP groups should be described/discussed.

4.      Why TNF-α-308 G>A (rs1800629) affected the levels of TNF-α should be discussed.

5.      In citation (first page), “J. Clin. Med. 202110, x.” should be changed to “J. Clin. Med. 202312, x.”.

6.      “Merck Millipore, MA, USA” (line 138) should be changed to “Merck Millipore, Burligton, MA”.

7.      “Nat Rev Cardiol. 2023 Mar 10” (Ref. 1) should be changed to “Nat Rev Cardiol 2023, 20, 560-573”.

8.      “Int J Mol Sci 2019; 20” (Ref. 17) should be changed to “Int J Mol Sci 2019; 20: 459”.

Round 2

Reviewer 2 Report

The authors transferred part of method into supplement. But still they keep the headings in the section of method similar and exact to that in their previously published paper in 2021?! No modification ? No change?!

Although some changes in the text and some new references have been added, but the authors insist to continue "copy and paste" approach of many paragraphs of their previous manuscript to the new one????.  For example: Fig. 1, Table !, Fig 2a....   WE already saw these items in your previous paper.

Please update and think about novelty.

Please authors, be direct in your presentation  of the data.:

1- Compare directly between your new data after 12 months CPAP and basic one.

2- No need to repeat the figures and results previously reported.

3- Discuss the clinical significance of assessment of TNF-apha polymorphism according to your new findings and how it helps clinicians to face the conditions of coronary artery diseases associated with OSA (i.e. clinical implementation of your new data in clinical field)

4- What prospective studies you suggest for more clarification of that clinical issue.

Reviewer 3 Report

Although most of points were suitably revised, numbers of points need clarifying and certain statements require further justification. These are given below.

1.      In Ref. 5, page numbers of “Coronary artery disease and sleep apnea” should be described and the ISBN number or DOI of “Principals and practice of sleep medicine. 2017” (978-0-323-24288-2, https://doi.org/10.1016/C2012-0-03543-0) should be added. In addition, the publisher name (maybe Elsevier, Philadelphia, PA) should be added.

2.      In Ref. 33, “Genet Mol Res. Aug 29 2016;15(3)” should be changed to “Genet Mol Res 2016; 13(3), gmr.15037292”. 

3.      In Ref. 45, the journal name, volume, and pages should be “J. Clin. Lan. Anal. 2018; 32(1), e22153”.

4.      In Ref. 49, “Sleep. 2017;40(11)” should be changed to “Sleep. 2017;40(11), zsx157”.
